# Rapid Multicomponent Alloy Solidification with Allowance for the Local Nonequilibrium and Cross-Diffusion Effects

**DOI:** 10.3390/ma16041622

**Published:** 2023-02-15

**Authors:** Sergey L. Sobolev, Mikhail G. Tokmachev, Yuri R. Kolobov

**Affiliations:** 1Federal Research Center of Problems of Chemical Physics and Medicinal Chemistry, Russian Academy of Sciences, 142432 Chernogolovka, Russia; 2Department of Applied Mathematics, Tikhonov Moscow Institute of Electronics and Mathematics, National Research University “Higher School of Economics”, 123458 Moscow, Russia

**Keywords:** mathematical modeling, metallic alloys, multi-component diffusion, ternary systems, numerical calculations

## Abstract

Motivated by the fast development of various additive manufacturing technologies, we consider a mathematical model of re-solidification of multicomponent metal alloys, which takes place after ultrashort (femtosecond) pulse laser melting of a metal surface. The re-solidification occurs under highly nonequilibrium conditions when solutes diffusion in the bulk liquid cannot be described by the classical diffusion equation of parabolic type (Fick law) but is governed by diffusion equation of hyperbolic type. In addition, the model takes into account diffusive interaction between different solutes (nonzero off-diagonal terms of the diffusion matrix). Numerical simulations demonstrate that there are three main re-solidification regimes, namely, purely diffusion-controlled with solute partition at the interface, partly diffusion-controlled with weak partition, and purely diffusionless and partitionless. The type of the regime governs the final composition of the re-solidified material, and, hence, may serve as one of the main tools to design materials with desirable properties. This implies that the model is expected to be useful in evaluating the most effective re-solidification regime to guide the optimization of additive manufacturing processing parameters and alloys design.

## 1. Introduction

Rapid solidification of multicomponent alloys (three or more components) is pertinent to many commercial materials and industrial processes such as casting, welding, 3D printing, and additive manufacturing [1,2,3,4]. The additive manufacturing is a new and advanced technology where the final 3D products are produced by the layer-upon-layer joining of the material. The high technological potential of the 3D printing and additive manufacturing has attracted much attention to rapid multicomponent alloy solidification and raised challenging questions from a fundamental point of view, which have been discussed by many authors [5,6,7,8,9,10,11,12,13,14,15,16,17].

Ludwig [5] has developed a theory for multicomponent solute trapping, which enables the determination of the *n* solid concentrations by solving an *n*-dimensional nonlinear system of equations, called first response functions. Hunziker [6] obtained analytical solutions for the solute diffusion fields during plane front and dendritic growth in multicomponent alloys, considering the diffusive interaction between the species, and examined the stability of solid–liquid interfaces. Altieri and Davis [7] studded linear stability analysis of multi-component alloys using one-sided diffusion and frozen temperature approximations and demonstrated that the multi-component systems are, in general, less stable than binary ones. Yu et al. [8] reported experimental results on the composition and crystallography of oxides formed on NiCrMo alloys during both high-temperature oxidation and aqueous corrosion experiments. Using a theory for nonequilibrium solute capture that combines thermodynamic, kinetic, and density functional theory analyses, the authors [8] demonstrated that the composition and crystallography are controlled by the rapidly moving interface. Wang and Liu [9] reviewed the basic philosophy for the phenomenological irreversible thermodynamics, the methods to obtain the governing equations for the evolution of multicomponent solidifying systems and the potential applications to other metallurgical phenomena. Xiong et al. [10] used a phase–field model with finite interface dissipation to construct kinetic phase diagrams in the ternary Al-Cu-Li system. Based on the predicted kinetic phase diagrams, it was found that with the increase in interface velocity and/or temperature, the gap between the liquidus and solidus gradually reduces, which illustrates the effect of solute trapping and tendency of diffusionless solidification. To predict microsegregation in the single-crystal superalloys of laser rapid directional solidification, Liang et al. [11] developed a microsegregation model for rapid solidification of multicomponent alloys considering nonequilibrium solute redistribution and dendritic tip undercooling. Mohan and Phanikumar [12] studied experimentally and theoretically rapid solidification of undercooled Ni–Fe–Si alloy system. Undercooling experiments were performed using flux encapsulation along with in situ measurement of recalescence speed and the highest growth velocity achieved was 30.4 ± 2.2 m/s, whereas phase–field simulations were based on a classical local equilibrium diffusion equation of parabolic type [12]. Dudorov and Roshchin [13] described a crystal growth process from a multicomponent melt using a new variation approach, which considers the interaction between thermal and diffusion processes, and constructed a metastable phase diagram. Sobolev et al. [14] suggested and analyzed a steady-state model of rapid solidification of multicomponent alloys with allowance for the cross-diffusive interaction between different solutes and the local nonequilibrium diffusion effects in the bulk liquid. In ternary systems the model predicts two-step transition to diffusionless solidification: from diffusion-controlled solidification of both solutes at low interface velocity to diffusion-controlled solidification of one solute and diffusionless solidification of another solute at intermediate interface velocity and to purely diffusionless solidification at high interface velocity [14]. Microstructures, solid–liquid interface morphologies and preferential orientations of the Fe-Al-Ta eutectic composites at different solidification rates were studied by Cui et al. [15]. Wang et al. [16] studied phase equilibrium of the Co-Ta-Si ternary system, which provided the key experimental data for the establishment of Co-based superalloys thermodynamic database.

Additive manufacturing of metallic alloys is a rapid solidification process when the solid–liquid interfaces may move with velocities as large as 1–30 m/s (see, for example [12,18,19], and references therein), which can be reached during re-solidification after ultrashort pulse laser melting [20,21,22,23,24]. At such velocities, solute diffusion in the liquid phase occurs under far from local equilibrium conditions and classical diffusion equation of parabolic type is not adequate for describing space–time evolution of solute concentrations [14,17,18,19]. To consider the deviation from local equilibrium of solute diffusion the liquid phase under rapid solidification conditions, the local nonequilibrium diffusion model (LNDM) [25,26,27] has been introduced by using diffusion equation of hyperbolic type, which is not based on the local equilibrium assumption. It has been demonstrated that the local nonequilibrium effects change drastically the diffusion field in the bulk liquid leading to a sharp transition from diffusion-controlled growth to kinetic-controlled growth with complete solute trapping [25,26,27]. The transition occurs when the interface velocity V passes through the critical point $V=V^{D}$, where $V^{D}$ is the solute diffusive velocity, i.e., velocity of propagation of diffusive disturbances. This implies that the transition to diffusionless and partitionless solidification is a purely diffusive phenomenon—it occurs independently of the interface solidification kinetics as soon as the interface velocity V reaches $V^{D}$ [25,26,27]. The LNDM has been also used to study various phenomena pertinent to rapid alloy solidification, for example, nonequilibrium kinetic phase diagrams [10,28], dendrite solidification [29,30], transient effects [19,31]. Moreover, the LNDM has been used in combination with other approaches such as cellular automata [30], discrete model [32], effective mobility approach [33], non-local approach [18,34], phase-field model [35,36], molecular dynamics, and Monte Carlo simulations [37].

Another important feature of multicomponent alloy solidification is the diffusional interaction between the species or cross-diffusion effects when diffusion fluxes of each component depend on the concentration gradient of all the components [14,17,38]. Off-diagonal diffusion terms of the diffusion matrix are usually neglected for multicomponent liquids, yet there is little justification, because in some situations the diffusional interaction between the species in multicomponent alloys plays an important role but the effect of the off-diagonal diffusion terms on diffusional transformation kinetics in multicomponent systems is still an open question.

Transient effects play an important role in rapid alloy solidification under ultrashort laser irradiation because the process occurs on short space and time scales [21,35]. For example, a transient approach to rapid solidification of SiAs alloy predicts that the interface temperature increases with decreasing interface velocity in the whole range of experimental parameters, which corresponds to the experimental results, whereas a steady-state approach predicts the opposite dependence in the small rage of the interface velocity, where the transient effects play the most important role (see discussion in [19,31]).

The purpose of this work is to develop a physical and mathematical model of rapid curing of the surface layer of a multicomponent alloy melted under the influence of an ultrashort laser impulse. At the same time, the model must take into account diffusion interactions between particles (non-diagonal diffusion conditions), local nonequilibrium, and transient effects. The development of such an approach and modeling methods will provide a valuable basis for studying the fundamental properties of highly nonequilibrium solidification of multicomponent alloys and will lead to more realistic forecasts of real processes, which will reduce the need for experimental work aimed at controlling the quality and properties of the materials of the final product.

## 2. General Model (*N*-Component System)

When a pulsed energy source, for example, laser or focused electron beam, irradiates the surface of a metal, the energy is absorbed, and melting may be induced near the surface. The melting is followed by the rapid re-solidification. Today, ultrashort energy sources allow rapid melting of layers as thin as several nanometers with re-solidification occurring on a time scale of picoseconds [39]. The corresponding solid–liquid interface velocity during re-solidification may exceed 1 m/s, which significantly suppresses the equilibrium partitioning of the species [19,25,26,27,30]. Figure 1 schematically shows the moving interface during re-solidification of a thin metal layer after ultrashort pulse laser melting.

### 2.1. Solute Diffusion in the Liquid Phase

Multicomponent diffusion is described by the Fick law for an *n*-component system, which in 1D takes the form [6,14,38]
(1)$$J_{i}=-\sum_{j=1}^{N-1}D_{ij}\frac{\partial C_{i}}{\partial x}$$
where $J_{i}$ is the flux of the component i, $C_{i}$ is the concentration of the component i, $D_{ij}$ are diffusion coefficients, which constitute an (N−1)×(N−1) diffusion matrix, and x- spatial coordinate. In general, the diffusion coefficients are not symmetric $\left(D_{ij}\neq D_{ji}\right)$. The diagonal terms $D_{ii}$ are so-called the “main terms” of the diffusion matrix, because they are commonly larger than the off-diagonal terms $D_{ij}$ and similar in magnitude to the binary values. The off-diagonal terms are usually ten percent or less of the on-diagonal terms.

Depending on the laser excitation strength, melting occurs roughly on a picosecond timescale followed by re-solidification with fast moving solid–liquid interface. In such a case solute diffusion in the bulk liquid occurs under local nonequilibrium diffusion conditions [18,19,25,26,27,31,32] and the generalization of the classical Fick law with allowance for the diffusional interaction between different solutes is given by [14,18]
(2)$$J_{i}+\tau_{i}\frac{\partial J_{i}}{\partial t}=-\sum_{j=1}^{N-1}D_{ij}\frac{\partial C_{i}}{\partial x}$$
where *t* is time and τ is the relaxation time of the diffusion flux of component *i*.

Combining Equation (2) with the mass conservation law for corresponding solute, we obtain the local nonequilibrium solute diffusion equations for the multicomponent system in the form [14,18]
(3)$$\frac{\partial C_{i}}{\partial t}+\tau_{i}\frac{\partial^{2}C_{i}}{\partial t^{2}}=-\sum_{j=1}^{N-1}D_{ij}\frac{\partial C_{i}}{\partial x}$$

When the off-diagonal terms $D_{ij}=0$, Equation (3) are of hyperbolic type and describe the transmission of concentration disturbances with finite velocities $V_{i}^{D}=\sqrt{\frac{D_{ii}}{\tau_{i}}}$, which are usually called “diffusive velocities” [18,19,25,26,27,31,32]. 

### 2.2. Interface Velocity

In one dimension (1D) solid-liquid interface velocity *V* during alloy solidification is given by [38,39]
(4)$$V\left(t\right)=\frac{dz}{dt}=V_{0}\left(T\right)\left[1-e^{-\frac{\Delta G}{RT}}\right]$$
where z is the coordinate of the interface, T is the interface temperature, ΔG is the difference between the Gibbs free energy of solid and liquid responsible for interface motion (driving force for solidification), $V_{0}$ is the maximal growth velocity when the driving force is infinite (no backward hopping).

### 2.3. Gibbs Free Energy

In general, the Gibbs free energy is given by
(5)$$G=G_{ref}+G_{mix}+G_{cros}+G_{neq}$$
where $G_{ref}$ is the reference (standard) term, $G_{mix}$ is the mixing term, $G_{cros}$ is the term due to interaction between the solutes, and $G_{neq}$ is the nonequilibrium term due to local nonequilibrium diffusion effects, respectively. The reference term corresponds to a simple mechanical mixture of the Gibbs energy of the constituent components and is given by [38,39]
(6)$$G_{ref}=\sum_{i=1}^{N}C_{i}G_{i}^{0}\left(T\right)$$

Here $C_{i}$ is the molar fraction of component i, ${}_{i}G_{i}^{0}\left(T\right)$ is the standard Gibbs free energy of component i at the temperature T. The ideal mixing term corresponds to the entropy of mixing for an ideal solution [38,39]:(7)$$G_{mix}=RT\sum_{i=1}^{N}C_{i}\ln C_{i}$$

The excess term takes into account the deviations form ideal mixing that are due to specific interactions between the components [38]:(8)$$G_{cros}=\sum_{i=1}^{N-1}\sum_{j=i+1}^{N}C_{i}C_{j}L_{i:j}$$

Here, $L_{i:j}$ are pairwise interaction parameters that can be temperature dependent [38]. 

The local nonequilibrium correction to the Gibbs free energy change, which arise during rapid alloy solidification when the solid-liquid interface moves with velocity of the order of the diffusive velocity, is given by [31,40]
(9)$$G_{neq}=\frac{1}{2}RT{\sum_{i=1}^{N-1}C_{i}^{S}\frac{{\left(1-K_{i}\left(V\right)\right)}^{2}}{K_{i}\left(V\right)}\left(\frac{V}{V_{i}^{D}}\right)}^{2}$$

## 3. Ternary System

### 3.1. Equations

We focus on one-dimensional description, which is a reasonable approximation when the laser spot size is much larger than the depth melted by the irradiation. That is typical for ultrashort-pulse laser melting experiments. For the ternary system, Equation (3) is reduced to [14,18]
(10)$$\frac{\partial C_{1}}{\partial t}+\tau_{1}\frac{\partial^{2}C_{1}}{\partial t^{2}}=D_{1}\frac{\partial^{2}C_{1}}{\partial x^{2}}+D_{12}\frac{\partial^{2}C_{2}}{\partial x^{2}}$$
(11)$$\frac{\partial C_{2}}{\partial t}+\tau_{2}\frac{\partial^{2}C_{2}}{\partial t^{2}}=D_{21}\frac{\partial^{2}C_{1}}{\partial x^{2}}+D_{2}\frac{\partial^{2}C_{2}}{\partial x^{2}}$$

A solute partition coefficient with allowance for the local nonequilibrium diffusion effects are given by (*i* = 1, 2) [14,18,25,26,27]
(12)$$K_{i}\left(V\right)={\frac{C_{i}^{sol}}{C_{i}^{liq}}|}_{x=z}=\left\{ \begin{array}{l} \frac{K_{i}^{E}\left(1-{\left(\frac{V\left(t\right)}{V_{i}^{D}}\right)}^{2}\right)+\frac{V\left(t\right)}{\gamma_{i}}}{\left(1-{\left(\frac{V\left(t\right)}{V_{i}^{D}}\right)}^{2}\right)+\frac{V\left(t\right)}{\gamma_{i}}},V<V_{i}^{D},\\ 1,V>V_{i}^{D}, \end{array}\right.$$
where $V_{i}^{D}=\sqrt{\frac{D_{i}}{\tau_{i}}}$ is the diffusive velocity in liquid phase, $K_{i}^{E}$ is the equilibrium partition coefficient of component *i*; $\gamma_{i}$ is the kinetic coefficient of component *i*. 

### 3.2. Initial and Boundary Conditions

Initial and boundary conditions for the ternary system are given by
(13)$${C_{i}^{sol}|}_{t=0}={C_{i}^{liq}|}_{t=0}=C_{i}^{0},i=1,2,3,$$
(14)$${\frac{\partial C_{1}^{liq}}{\partial t}|}_{t=0}={\frac{\partial C_{2}^{liq}}{\partial t}|}_{t=0}=0,$$
(15)$${\frac{\partial C_{1}^{liq}}{\partial x}|}_{x=l}={\frac{\partial C_{2}^{liq}}{\partial x}|}_{x=l}=0,$$
(16)$${\frac{C_{i}^{sol}}{C_{i}^{liq}}|}_{x=z}=K_{i}\left(V\right),i=1,2,$$
(17)$$\int_{0}^{z}C_{i}^{sol}\left(x\right)dx+\int_{z}^{l}C_{i}^{liq}\left(x\right)dx=C_{i}^{0}\cdot l,i=1,2,$$
(18)$$z\left(t\right)=\int_{0}^{t}V\left(t\right)dt,z\left(0\right)=0.$$
where x=z(t) is the coordinate of the moving solid–liquid interface (see Figure 1); at x>z(t)—liquid (*liq*) phase; at x<z(t)—solid (sol) phase. Equations (13) and (14) are the initial conditions, which define initial concentrations and their time derivatives in the liquid phase 0<x<l. Equation (15) implies that there are no solute fluxes in the environment at x=l. Equation (16) describes solute redistribution at the moving solid–liquid interface at x=z(t), where the solute partition coefficients $K_{i}\left(V\right)$ are given by Equation (12). Equation (17) is nothing else but the mass balance law. Equation (18) defines the position of the solid–liquid interface moving with velocity *V* given by Equation (4).

### 3.3. Calculations 

For the numerical solution of Equations (10) and (11), their difference analogues were used in accordance with the template shown in Figure 2a. The difference scheme is implicit, which means it is absolutely stable. To find the values of the desired functions on a new time layer (marked with the index m+1 in Figure 2a) we obtain a system of linear equations (in accordance with the indicated three-point pattern). To solve this system, we developed an economical numerical algorithm based on the generalization of the tridiagonal matrix one.

It should be noted that the computational domain shrinks over time since the task has a moving interface. Thus, at each time step, the grid points change their position (see Figure 2b). Therefore, before calculating the values of the desired functions at a new time layer, it was necessary to estimate the values of the functions on the current and previous time layers at new points. The estimation was carried out by interpolation with natural cubic splines.

## 4. Results and Discussion

In this section we present results of the numerical simulation of the ternary alloy re-solidification described by Equations (10)–(18). The calculations were carried out with following parameters: initial length of the molten layer l = 500 nm, process temperature T = 1500 K, interface velocity $V_{0}$ = 3 m/s. For the sake of simplicity, we also assume that $L_{i:j}=0$ in Equation (8). All other parameters needed for numerical calculations are summarized in Table 1.

In Figure 3 we show solute concentration profiles in the liquid and solid phases as functions of the coordinate x at $V<V_{i}^{D}$ without cross diffusion effects. As expected, in such a case solidification occurred in the diffusion-controlled regime with solute partitioning at the interface, which was manifested by the two moving concentration spikes at the interface [14]. As the solid–liquid interface reached the end of the liquid phase (x=l in Figure 1), the boundary effects began to play an important role and the solute concentrations in the solid phase significantly increased near the boundary (see Figure 3c). 

Solute concentration profiles in the liquid and solid phases are shown in Figure 4 as functions of the coordinate x at $V<V_{1}^{D}$ and $V>V_{2}^{D}$ without cross diffusion effects. In this case, during the re-solidification we observed the concentration spike of solute 1, whereas the concentration of the solute 2 did not change at the interface (see Figure 4a,b). This implies that, at the interface, solidification occurred in the diffusion-controlled regime with partitioning regarding solute 1, while solute 2 solidified in the diffusionless and partitionless regime. After complete re-solidification, the concentration of solute 1 significantly increased near the boundary, whereas the concentration of solute 2 did not change (see Figure 4c).

The influence of the strong positive cross diffusion effect on solute concentration profiles in the liquid and solid phases is shown in Figure 5 at $V<V_{1}^{D}$ and $V>V_{2}^{D}$. In this case, the cross diffusion significantly affected the solute concentration profiles not only at the final stage when complete re-solidification occurred (see Figure 5c), but also far from the boundaries (Figure 5a,b). In spite of the fact that the effective diffusion coefficient of the solute 2 was zero because the solidification occurred at $V>V_{2}^{D}$ [14,18,25,26,27], the concentration gradient of solute 1 led to the non-zero diffusion flux of solute 2 due to the cross diffusion effects (see Equations (1) and (2)). That resulted in non-zero concentration gradient of solute 2 and, consequently, a corresponding spike in the concentration of solute 2 at the interface (see Figure 5a,b). After complete re-solidification (see Figure 5c), the solute concentrations increased near the surface of the alloy.

Figure 6 shows solute concentration profiles in the liquid and solid phases as functions of the coordinate x at $V<V_{1}^{D}$ and $V>V_{2}^{D}$ with strong negative cross diffusion effect. As in the previous case, the effective diffusion coefficient of the solute 2 was zero, but strong cross diffusion effect led to the non-zero diffusion flux and concentration gradient of solute 2, which resulted in the negative concentration spike of solute 2 at the interface (see Figure 6a,b). Note that, after complete re-solidification, the concentration of solute 2 in the solid phase decreased in the vicinity of the metal surface in comparison with its bulk value (see Figure 6c). 

Calculations demonstrate that, except for the initial and final transient, the interface velocity V slightly depended on time.

## 5. Conclusions

In this paper we consider a model of rapid multicomponent alloy solidification with allowance for the local nonequilibrium and cross-diffusion effects in the liquid phase, which arise on the ultrashort space and time scale. The model successfully describes not only steady-state, but also initial and final transient stages, thus providing a further basis for the understanding of the multi-component alloy solidification. Numerical simulations demonstrate that, depending on the local nonequilibrium and cross-diffusion effects, the re-solidification may occur in three main regimes, namely, purely diffusion-controlled with solute partition at the interface, partly diffusion-controlled with weak partition, and purely diffusionless and partitionless. The type of the regime governs the final composition and, hence, physical properties of the re-solidified material, which may serve as one of the main tools to design materials with desirable properties. This implies that the model is expected to be useful in evaluating the most effective re-solidification regime to guide the optimization of additive manufacturing processing parameters and multi-component alloys design.

## Figures and Tables

**Figure 1 materials-16-01622-f001:**
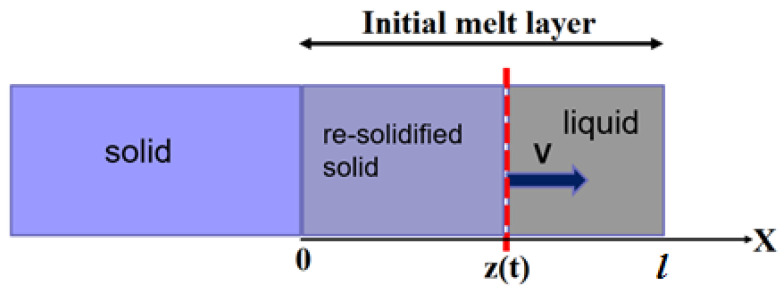
Schematic of an advancing solid–liquid interface during re-solidification of a thin metal layer after ultrashort pulse laser melting. l—is the initial thickness of the melted layer. Liquid phase (*liq*) is depicted at x>z(t) and solid phase (*sol*) is depicted at *x* < *z*(*t*).

**Figure 2 materials-16-01622-f002:**
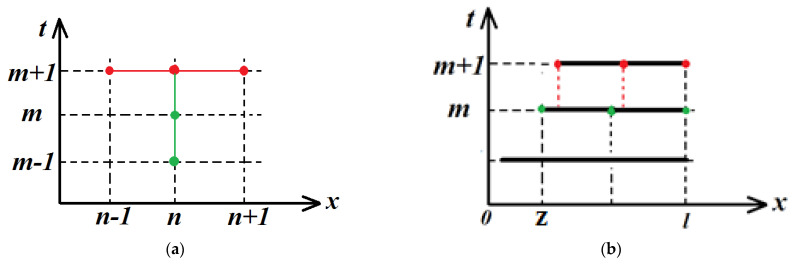
(**a**) difference scheme template, (**b**) illustration of the interpolation procedure. The red nodes of the grid are located on a new time layer, the values of the functions in them are to be determined. Green nodes are on the current or past time layer, the values of the functions in them are already known.

**Figure 3 materials-16-01622-f003:**
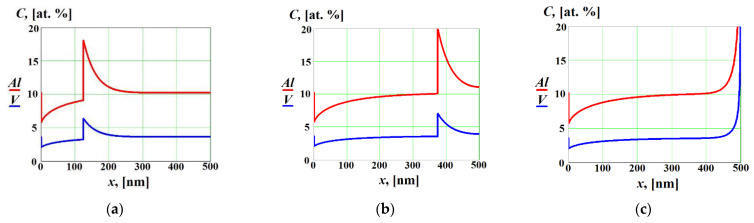
Solute concentrations in the liquid and solid phases as functions of the coordinate x at $V<V_{i}^{D}$ without cross diffusion effects ($D_{12}=D_{21}=0$) at different time moments: (**a**) t=55 ns, (**b**) t=163 ns, (**c**) t=217 ns (complete re-solidification).

**Figure 4 materials-16-01622-f004:**
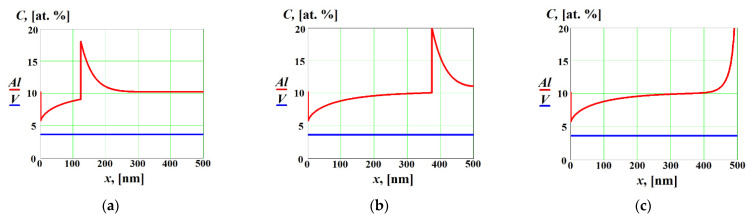
Solute concentrations in the liquid and solid phases as functions of the coordinate x at $V<V_{1}^{D}$ and $V>V_{2}^{D}$ without cross diffusion effects ($D_{12}=D_{21}=0$) at different time moments: (**a**) t=54 ns; (**b**) t=163 ns; (**c**) *t* = 217 ns (complete re-solidification).

**Figure 5 materials-16-01622-f005:**
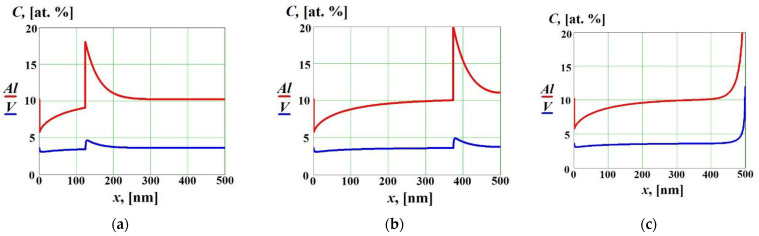
Solute concentrations in the liquid and solid phases as functions of the coordinate *x* at $V<V_{1}^{D}$ and $V>V_{2}^{D}$ with strong positive cross diffusion effects ($D_{12}=0$, $D_{21}=1.5\cdot 10^{-8}$ m^2^/s) at different time moments: (**a**) t=54 ns; (**b**) t=163 ns; (**c**) *t* = 217 ns (complete re-solidification).

**Figure 6 materials-16-01622-f006:**
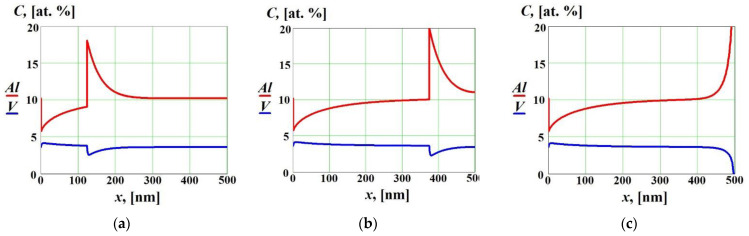
Solute concentrations in the liquid and solid phases as functions of the coordinate *x* at $V<V_{1}^{D}$ and $V>V_{2}^{D}$ with strong negative cross diffusion effects ($D_{12}=0$, $D_{21}=-1.5\cdot 10^{-8}$ m^2^/s) at different time moments: (**a**) t=54 ns; (**b**) t=163 ns; (**c**) *t* = 217 ns (complete re-solidification).

**Table 1 materials-16-01622-t001:** Parameters used for calculation.

	Component 1	Component 2	Component 3
Component title	Al	V	Ti
Mol. weight [g/mol]	27	51	48
C_0_ [mass. %]	6	4	90
$G_{i,liq}^{0}$ [kJ/mol]	223.3	210.6	218.8
$G_{i,sol}^{0}$ [kJ/mol]	200	200	200
$D_{i}$, [10^−9^ m^2^/s]	100	5	
τ, [ns]	1	1	
$K_{i}^{E}$, [-]	0.1	0.1	
$\gamma_{i}$, [m/s]	3	1	

## Data Availability

All data regarding the simulation and modeling are available on request.

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
