# Peer review of "Rapid Multicomponent Alloy Solidification with Allowance for the Local Nonequilibrium and Cross-Diffusion Effects"

_materials, 2023, doi:10.3390/ma16041622_

Round 1
Reviewer 1 Report
The research article entitled “Laser Irradiation: Local Nonequilibrium Diffusion Model with Allowance for the Interaction Between the Species” is a novel work and is having publishable information. The authors have developed a model suitable for multi-component alloy processed by additive manufacturing. It can be considered for its publications. I have observed the following points which the authors have addressed on it.
1. In abstract, the authors have mentioned the application of their model to multi-component metal alloys. Whether it can be applicable low-entropy, medium entropy alloys as-well or only high-entropy alloy alone. It has to be addressed. The authors have applied to Ti-Al-V ternary alloy. Here, what is the composition of this system used either equal molar ratio or non-equal molar ratio. It is not clear in abstract.
2. Why the authors have not used the word of entropy alloys instead they used multi-component? It has to be checked and addressed in introduction part, entire manuscript, and in the title as-well
3. In Table 1, some values might have taken from some sources. References may be incorporated here. How the authors have taken these values? It has to be addressed
4. Similarly, the authors have taken some initial values (molten layer as 500 nm, process temperature as 1500K, and interface velocity as 3m/s). How it was selected? Need to cite the selected source.
5. What is the accuracy of developed model like the correlation coefficient or regression coefficient?
Reviewer 2 Report
Dear authors, I consider that your manuscript needs a revision. Please see the remarks presented in the attached review document.

Reviewer 3 Report
This manuscript used a nonequilibrium model to investigate the solute distribution during solidification of Ti6Al4V. Based on the model, the authors found that both local nonequilibrium and cross-diffusion have significant impact on the final solute distribution in the solid phase. The manuscript contains some interesting data, but has a number of significant weaknesses.
1. The equations used in this manuscript are already well established in previous studies. This manuscript brings very little new understanding into the field.
2. Many variables in equations are not defined, including Equation (2), (6), and (12).
3. Some property values used in calculation (in Table 1) are not correct. Where did authors get the partition coefficients of Al and V in Ti? The authors should add references for sources of these numbers.
4. Why are the Gibbs free energies for solid Al, V and Ti the same in Table 1?
5. Where did authors get the value for D21 in Figure 5?
6. Experimental results should be provided to verify the simulation results.
Round 2
Reviewer 1 Report
The authors have worked based on raised comments. Hence, I am recommend to accpet this article
Reviewer 2 Report
Dear authors, I have read your revised manuscript and I found it more sound than the previous version. You answered clearly and satisfactory to all my previous review remarks. I recommend publishing the paper in the MDPI journal as it is.
Reviewer 3 Report
The manuscript has been revised with satisfaction.